# Learning from stakeholders to inform good practice guidance on consent to research in intensive care units: a mixed-methods study

Katie Paddock [1,2] Kerry Woolfall,[2] Anna Kearney,[2] Natalie Pattison,[3,4] Lucy Frith [5] Carrol Gamble,[6] Ingeborg Welters [7,8] John Trinder,[9] Bridget Young[2]

For numbered affiliations see end of article.

**Correspondence to**
Dr Katie Paddock;
k.paddock@mmu.ac.uk

## ABSTRACT

**Objectives** Obtaining informed consent from patients in intensive care units (ICUs) prior to enrolment in a study is practically and ethically complex. Decisions about the participation of critically ill patients in research often involve substitute decision makers (SDMs), such as a patient's relatives or doctors. We explored the perspectives of different stakeholder groups towards these consent procedures.

**Design and methods** Mixed-methods study comprising surveys completed by ICU patients, their relatives and healthcare practitioners in 14 English ICUs, followed by qualitative interviews with a subset of survey participants. Empirical bioethics informed the analysis and synthesis of the data. Survey data were analysed using descriptive statistics of Likert responses, and analysis of interview data was informed by thematic reflective approaches.

**Results** Analysis included 1409 survey responses (ICU patients n=333, relatives n=488, healthcare practitioners n=588) and 60 interviews (ICU patients n=13, relatives n=30, healthcare practitioners n=17). Most agreed with relatives acting as SDMs based on the perception that relatives often know the patient well enough to reflect their views. While the practice of doctors serving as SDMs was supported by most survey respondents, a quarter (25%) disagreed. Views were more positive at interview and shifted markedly depending on particularities of the study. Participants also wanted reassurance that patient care was prioritised over research recruitment. Findings lend support for adaptations to consent procedures, including collaborative decision-making to correct misunderstandings of the implications of research for that patient. This empirical evidence is used to develop good practice guidance that is to be published separately.

**Conclusions** Participants largely supported existing consent procedures, but their perspectives on these consent procedures depended on their perceptions of what the research involved and the safeguards in place. Findings point to the importance of explaining clearly what safeguards are in place to protect the patient.

## INTRODUCTION

Clinical research in intensive care units (ICUs) is a vital aspect of medical practice,

## STRENGTHS AND LIMITATIONS OF THIS STUDY

⇒ This study explores the views and experiences of different stakeholder groups towards research recruitment and consent procedures for adults in the intensive care unit, a topic which has previously been little explored in the UK.

⇒ The use of mixed-methods provides more comprehensive knowledge than only using one approach to data gathering, which increases the study's impact and utility.

⇒ To keep surveys a manageable length, surveys did not include substantial demographic data.

⇒ We encouraged participating sites to complete screening logs to enable us to report survey response rates, but most sites did not have the staffing to support this.

⇒ While we sampled for diversity of views, by virtue of their participation in this study it is likely that most participants had favourable views about research, and so it is possible that views of stakeholders with opposing views may be under-represented.

but obtaining informed consent from patients prior to their enrolment in a study is often not possible and recruitment is therefore ethically complex. Though prospective patient informed consent is considered to be 'gold standard', a patient must be able to understand and retain the information on risks and benefits presented to them and clearly communicate their decision.[1–3] Yet only around 10% of ICU patients have capacity to provide informed consent.[4–9] The use of substitute decision makers (SDMs) to facilitate inclusion of incapacitated patients has been developed to balance the rights and welfare of the patient with the necessity of conducting research. Examples of SDMs include a patient's relative or next of kin. Also, in some circumstances, a doctor who is not involved with the study and is therefore

BMJ

considered to be independent, often referred to as a professional legal representative or nominated consultee. Implementing substitute decision making also has its challenges, particularly where there is a short time window for recruitment.[3 10] Family receptivity to discussions about research in ICUs may be affected by the often sudden hospitalisation, and many relatives will be preoccupied with dealing with the emotional impact of this. Furthermore, there are situations when even substitute decision making by a relative or independent practitioner is not feasible, such as when a study is investigating under emergency situations, and delaying administration of treatment to consult with SDMs could compromise patient outcomes. In such situations, patients can be enrolled in studies without prior consent, providing this is approved by an ethics committee.[11–14]

These challenges to seeking consent may compromise either efficient recruitment or (potentially) meeting patients' wishes. Despite this, little research has been conducted on the perspectives of stakeholders on recruitment and consent processes for adult ICU studies in Europe, and evidence is particularly lacking in the UK. In the absence of such evidence, investigators and Research Ethics Committees (RECs) have little choice but to base their judgements about enrolment processes on speculations and assumptions about how recruitment and consent might be experienced by patients and relatives.[15 16] Findings in non-ICU contexts have repeatedly indicated that such speculations can be mistaken, and pointed to how the perspectives of patients and relatives often diverge markedly from what study teams and RECs assume these to be.[17–23] Evidence on the perspectives of patients and other stakeholders is vital for improving consent processes, and such knowledge is gradually accumulating overseas, particularly in North America.[5 24–30] However, cultural, health system, and regulatory differences mean there is a need to investigate whether this overseas evidence has applicability to the UK. Though healthcare researchers will learn from and adjust how they approach patients and relatives about research through experience and observing others,[31] bad practices can also be taught/reinforced. Hence, it is important to ensure that consultation and consent procedures align with the perspectives of key stakeholder groups—patients, relatives and healthcare practitioners (HCPs), and that erroneous understandings are identified and corrected. The overall aim of our study—the Perspectives Study—was to provide evidence on the views and experiences of key stakeholders with relevant lived experience and use this evidence to develop good practice guidance for the recruitment and consent of patients in the ICU, including the involvement of SDMs.

## METHODS

We drew on a mixed-methods approach[32–35] involving four interlinked work streams and the integration of quantitative and qualitative evidence from ICU patients, their relatives and HCPs, which included doctors, nurses, allied health professionals and pharmacists. Details of workstream 1 are presented elsewhere.[36] Mixed methods provide more comprehensive knowledge than only using one approach[37] to data-gathering and we anticipated that this would help to increase the study's impact and utility.

### Patient and public involvement

This project was supported by an advisory group comprising former ICU patients, relatives of former patients, ICU clinicians and experienced ICU researchers. This advisory group informed the study design, including participant recruitment, survey and topic guide design, and analysis.

### Sampling and recruitment

Potentially eligible National Health Service (NHS) hospitals were identified by searching the UK National Institute for Health and Care Research Clinical Research Network portfolio, and through a call for expressions of interest via email invitation. Seventy-two hospitals initially expressed an interest in participating. We purposefully sampled hospitals in England to include variation in terms of: experience of hosting/conducting ICU studies; number of ongoing studies; number of ICU/high dependency unit beds; admissions per year; number of clinical staff; and location. As Health Research Authority (HRA) regulations governing recruitment and consent of incapacitated patients to research vary across the devolved UK nations, this study focused on ICUs in England.

We conducted a survey with ICU patients, their visitors and HCPs, which explored experiences and views of ICU research recruitment and consent process. A subset of survey respondents participated in semistructured interviews to discuss their responses to the survey and to explore their views in more detail. Eligibility criteria required for inclusion:

► 18 years of age or over.
► Capacity to give consent at time of data collection.
► Could complete the survey in English.
► Were a patient currently receiving, or who had recently received care in an ICU setting.
► The relative/next of kin/close friend of such a patient.
► Were a doctor or nurse currently based in an ICU setting (whether actively involved in research or not).

Research nurses at participating sites distributed paper questionnaires to patients, visitors (hereto referred to as relatives), and HCPs. Sites were asked to recruit individuals whether or not they had previously been approached about a clinical research study while in the ICU, although the importance of including those who had been approached about a study was emphasised. Sites were also asked to recruit HCPs to represent a range of both clinical roles and involvement in ICU research. Participants had the option of completing a postal or online survey. A freepost self-addressed envelope was provided for completed postal surveys to be returned to the Perspectives study team.

**Table 1** Interview sampling matrix

| Patients and relatives | HCPs |
|---|---|
| Experience of being approached about research | Experience of involvement in the conduct of research |
| Age (18–24; 25–34; 35–44; 45–54; 55–64; 65–74; 70+) | Age (18–24; 25–34; 35–44; 45–54; 55–64; 65–74) |
| Gender | Gender |
| Socioeconomic status* | Role in ICU |
| Range of views of ICU consent procedures determined by the survey | |
| Consented/declined participation in an ICU study | |

*Based on the English Index of Multiple Deprivation, which is obtained by entering participants' postcodes into the Consumer Data Research Centre website.[65] This ranks every area in England from most to least deprived. The deciles were derived from the ranks, and then we further grouped these into most deprived (1–3), average deprivation (4–7) and least deprived (8–10) https:// maps.cdrc.ac.uk/#/geodemographics/imde2019/default/BTTTF FT/10/–0.1500/51.5200/.
HCPs, healthcare practitioners; ICU, intensive care unit.

Participants indicated on the survey whether they were willing to be contacted to take part in an interview. Those who provided contact details were sampled for interview to represent a range of attributes illustrated in table 1, including residence in areas of high/low/middle deprivation, enabling us to explore the views and experiences of participants from a range of backgrounds. Priority was given to interviewing participants who had experience of research whether as a patient, relative or HCP. HCPs with a research role are hereto referred as healthcare practitioner researchers (HCPRs).

Sampling for the qualitative interviews was guided by the concept of 'information power', whereby sample size is dependent on factors such as the breadth of study aims, homogeneity of the sample and interview quality.[38] Sampling continued until the point of information power was reached.

## The survey

Patients, relatives and ICU HCPs completed different versions of the survey. Where possible each version included the same questions, with wording changed to reflect the respondents. Participants responded using a five-point Likert scale. The survey included a brief introduction to The Perspectives Study, a link to an online version and comprised three sections: (1) demographic information; (2) experiences of being approached about research (patients and relatives) or of being involved in conducting ICU research (HCPs) and (3) views on a variety of consent and recruitment scenarios. A final open-ended item invited participants to add further comments relating to their experiences or views on research in the ICU.

Survey items were informed by findings from a linked qualitative interview study of ICU researchers and patient and public involvement contributors,[36] relevant literature[39–42] and discussion with our advisory group. The surveys were piloted with patients, relatives, and HCPs across three participating hospitals, and refined in consultation with the wider study team.

## Interviews

Semistructured interviews were conducted by KP (a psychologist) and AK (a health research methodologist), both with experience in qualitative methods, between January 2017 and June 2019. Participants were given the option of a face-to-face, telephone or video conference interview. All interviews were audiorecorded, and transcripts were checked for accuracy and pseudoanonymised prior to analysis. Questions explored participants' survey responses in more detail and the reasoning behind their responses, with additional questions on issues such as discussing research with bereaved relatives and conducting research without prior consent (RWPC).

Patient and relative participants were offered a £25 shopping voucher after being interviewed to acknowledge their contribution to the study.

## Analysis

Survey data were entered into SPSS (V.25) and analysed using descriptive statistics. Likert responses were combined for strongly agree and agree, and strongly disagree and disagree to ascertain whether participants were broadly positive, negative or neutral in their views Analysis of interview data was broadly interpretive and informed by thematic reflective approaches.[43 44] We conducted the analysis at multiple levels, from line-by-line coding, to consideration of participants' narratives at a holistic level in order to ensure coherence and contextualisation. KP led the analysis in close consultation with BY and KW. KP read all interview transcripts multiple times, with BY, KW and other team members (which included individuals with expertise in qualitative methods, ethics, critical care and clinical trials) reading a sample of pseudoanonymised transcripts and discussing these to develop the analysis, which was further discussed with the advisory group. We used NVivo software (V.10) to assist with indexing and coding of qualitative data.

Empirical bioethics[45] informed the analysis and synthesis of the data. KP cross-referenced themes identified from the qualitative analysis with related data from the survey analysis. In collaboration with the wider study team, KP and LF, a bioethicist, used ethical theory, principles and concepts such as autonomy, non-maleficence and therapeutic misconception to elucidate and analyse data. This was then used to draw normative conclusions that informed the development of our good practice guidance.[46] This approach places ethical issues in their social context, and enabled us to explore stakeholders' accounts of what they did in practice, and what good practice looked like for them. Empirical ethics has been

used in other studies to integrate empirical evidence and bioethical literature and draw practice-orientated conclusions.[47]

Further interviews were conducted where there was a divergence between qualitative and quantitative findings on the same issue, for example, where participants expressed negative views on a consent process on the survey but were more positive about this same process at interview.

In this paper, we focus on stakeholders' views on consent procedures. The survey and interviews also investigated participants' experiences of being approached about participating in research (patients and relatives) or conducting research (HCPs). These findings will be reported elsewhere. The full dataset and transcripts are available via the UK DataService repository.[48]

## RESULTS

We note that the terminology used in guidance and regulations on SDM processes in ICU studies often differs to that used by the public. For example, UK HRA guidance on clinical trials of medicinal drugs/devices (known as CTIMPS) with adults who are unable to consent for themselves, refers to seeking consent from a personal legal representative (usually a relative) or a professional legal representative (a doctor independent of the study). For other intrusive research (known as non-CTIMPS), researchers should consult with personal consultees (usually relatives) or nominated consultees (again, such as doctors independent of the study) for advice on whether the patient would wish to be included in a study, rather than seeking their consent. This means that the locus of responsibility for inclusion lies with the recruiting researcher, not the consultee. Representatives and consultees are collectively referred to as SDMs.[11–14] The legal arrangements regarding SDMs differs between countries. Reflecting the terminology more widely understood by the public, we generally used the term 'consent' and 'consultees' in the surveys and interviews to refer to the SDM process, and we therefore use this in describing the findings.

## Participants

Of 1453 returned surveys, 1409 were included in the analysis (ICU patients n=333, relatives n=488, HCPs n=588). Of all survey participants, 38% (n=532) indicated that they had been approached about research (patients n=115, relatives n=157) or were involved in conducting research (HCPs n=260) while in the ICU. A further breakdown of this sample and reasons for exclusion are presented in online supplemental file 1.

Five hundred and forty-five survey participants (38%) indicated that they would be happy to be approached about participating in an interview. We contacted 269 individuals for interview based on the aforementioned sampling matrix. The email address or phone number for 13 were incorrect, 3 declined to participate and 199

did not respond. We interviewed 60 participants (patients n=13, relatives n=30 and health practitioners n=17). Fifty-four participants interviewed had completed the survey while the remaining six were related to or lived with the participants who had completed the survey, and wished to be involved in an interview. This resulted in 49 one-to-one interviews and 11 group interviews.

Six interviewed patients had direct experience of being approached to consent for a study. A further 19 relatives interviewed recalled having been approached about research on behalf of, or with, a patient. Thirteen HCPs indicated that they had been involved in consent and/or recruitment to ICU studies. Online supplemental file 2 summarises interview participant characteristics.

Interviews were conducted in person (n=39), or by phone/video conference (n=21), and lasted between 34 and 120 min (mean=77 min).

## The importance of research

Although the NHS was not explicitly mentioned in the survey items, patients and relatives expressed great trust in the NHS and a belief in the importance of research to inform and improve care, in both open-ended survey responses and in interviews. Ninety per cent (301/329) of surveyed patients, 93% (452/484) of relatives and 98% (575/583) of HCPs agreed that clinical research in the ICU is important to help other patients in the future. Fifty-nine per cent (194/329) of surveyed patients and 53% (255/486) of relatives agreed that 'all ICU patients should take part in clinical research studies, unless a doctor advised against it'. Fewer surveyed HCPs (41%, 236/580) than patients and relatives agreed with this statement. In the interviews, several patients or relatives spoke positively of their or a relative's involvement in research at a previous point in their lives, or of having benefited from the knowledge that research can generate. Others used analogies based on their personal experience to illustrate the importance of research:

> I was a chef for many years and you can only develop a new recipe if you try adding a new ingredient and seeing if it makes it better.
>
> Patient 713

At interview, patients, relatives and HCPs who disagreed with this statement emphasised the importance of autonomy, took issue with the notion of 'all' patients being involved in research, and emphasised the importance of decisions about research participation being informed by patients' personal beliefs. These sentiments were particularly emphasised by doctors with research duties. Nevertheless, all participants acknowledged that critically ill patients are often not in a position to express their wishes at the point of recruitment.

## Timing the approach

In interviews, HCPRs stated that they aimed to speak to relatives about research at the earliest opportunity, while also ensuring the timing of this was appropriate. To

determine an appropriate time, they considered whether relatives were new to the ICU or had recently received, or were soon to receive, bad news about the patient's condition. All patients and relatives considered it important for HCPRs to update them about the patient's condition before being asked about research. The experience of most patients and relatives who had been approached about research was that the approach occurred sometime after their last update about the patient's condition, or very soon after being admitted. Interview participants commented that relatives and patients may not be well-inclined towards research if they are not first informed/updated about the patient's condition.

Relatives wanted to feel assured that the clinical considerations of the patient took precedence over research. A few participants reported that they had been approached about research as soon as they first entered the ICU ward, before they had had a chance to see the patient or receive an update on their condition and pointed to the difficulties this caused.

> We were both, you know, in a bit of a state…we wanted an update really…nobody else had approached us or, you know, we'd not spoken to anybody, we'd just arrived…we're there concerned about [name of patient] you know, is she going to live or die and they're saying well yes but meanwhile can I take some blood.
>
> Relative (group interview) 491

Many HCPRs spoke of aiming to provide such clinical information to patients and relatives before approaching them about research, having previously consulted the clinical care team in person, or first ensuring they are up to date with the patient's clinical status and progress. These consultations and notes helped HCPs to '[get] an understanding from the nurses looking after' a patient (HCP 404). They would sometimes offer insights about family dynamics or whether a relative was particularly upset, and so assisted HCPs in determining when and who to speak with about being involved in a research study:

> You get a feel when you look at the notes, etc, and you can usually get a feel, you think this is not going to go well, the sort of family circumstances … you kind of often go with that nurses' spidey sense. I'm like, well, I don't think this is gonna be right
>
> HCP 4148

However, HCPs also suggested that procedures across hospitals and individual clinicians differed as to whether updates on the patients' condition or family dynamics were provided before approaching patients and relatives about research. For some HCPRs, being independent of the patients' clinical care team coupled with the fast-paced nature of ICU research and the ICU environment often meant there was insufficient time, or it was not feasible to, co-ordinate with the clinical team before approaching the patient or family. Additionally, HCPRs differed in their expertise, and some believed it was not appropriate for them to provide detailed updates about the patient's condition when approaching relatives about research:

> If they have more questions about [the patient's condition], I would answer them within the sphere of competence that I have but, um, being very mindful of the limitations on what's appropriate for me to be discussing and under what circumstances they should really be talking to their medical team on the ward and while they're on the ward…
>
> HCP 351

During interviews, HCPRs were asked about consent-seeking and how they formally assessed capacity, such as if they used an assessment tool. Most described assessing capacity informally based on their interactions with the patient over time, typically over a day or more. Some also consulted with the clinical care team and relatives, though as discussed above, this was not always possible.

Several patients commented that they were approached about research when they were still finding it difficult to take in information or felt that they lacked the capacity to make an informed decision.

> I was able to make decisions myself, but it, it was just a case of not everything really sunk in as well as it could have done sort of thing, but, but that wasn't her fault. That was my fault.
>
> Patient 1417

### Framing the approach about research

For many interviewed patients and relatives, it was important that HCPRs clarified that they are acting in a research capacity when introducing themselves, as being approached by any HCP in the ICU can cause concern. As we also describe below, many patients and relatives were confused about the distinctions between research and clinical care. To help convey the distinction between research and care, some interviewees suggested that bedside nurses or other staff known to the family could introduce the researcher who intends to discuss a study. This could also help to present research as a collaborative effort in the ICU.

A few patients and relatives felt that HCPRs in the ICU could have done more to acknowledge the difficulty of the situation when either providing an update on the patient's care or enquiring about research. They wanted HCPRs to show empathy for their situation, as well as giving priority to the clinical care of the patient.

> Timing and having someone who's got a good bedside manner, not towards just the patient but that empathy of what the family is going through…you're just turning up fragile and angry at the world… someone could have a great bedside manner and still get it wrong.
>
> Relative 144

## Acceptability of personal consultees

When a patient is too ill to consent for themselves to participate in research, most surveyed patients (68% 224/329), relatives (83% 404/486) and HCPs (76% 445/584) considered it acceptable for relatives to consent on behalf of the patient who was to be involved in research. For many HCPs, this was because they trusted the processes surrounding the design, approval and conduct of research and 'the honesty and the veracity of any of the principal investigators' (HCP 102) to design a safe and rigorous study. During interviews, participants agreed that though it was more desirable to seek consent from patients in the first instance, when this was not possible, consent from a relative was often considered preferable to not enrolling a patient in to research. They linked this viewpoint to the beliefs that research is important to improve care. Most patients and relatives also felt that they would want to enable research to occur, or avoid impeding it in some way, and expressed confidence in relatives knowing the patient's preferences regarding research participation.

> Oh, I'd agree. Um, my family'll know that, you know, god, whatever, yeah, I'll go for it 'cause a) it might help me and b) it might help somebody else further on down the line so yeah, you know.
>
> Patient 713

A minority of patients and relatives differed slightly in this view, with some arguing that a relative should not consent on behalf of a patient to an invasive study.

> …I think it depends on what the research is, but I think if it's something that's non-invasive, then that's, that would be acceptable
>
> Patient 340

Others emphasised the importance of collaborating as a family to make decisions that represent the patient's wishes, and to alleviate the emotional burden on a single relative.

> I think my eldest sister is down as the next of kin, but when any decisions needed making, we're all equal… it was a, a decision that we all make. She wouldn't just say, do this, because obviously we all a-, have our own relationship with me mum, and know different things, so it's a discussion if one of us disagreed, then we'd have to speak about it.
>
> Patient 228

## Acceptability of consent from a doctor

As noted above, for non-CTIMPs, nominated consultees provide advice regarding the recruitment of an incapacitated patient to research, not consent on their behalf as they would in CTIMPs.[11] We used the more widely understood terminology of 'consent' to refer to in the survey and interviews.

The practice of doctors consenting to research on behalf of incapacitated patients was supported by most surveyed stakeholders regarding situations where a patient had no

known relatives (patients 60%, 197/328, relatives, 63%, 307/485, practitioners 63%, 368/582) or if time was too short to contact relatives (patients 57%, 188/329, relatives, 51%, 246/485, practitioners 53%, 309/583). Many patients (55%, 179/328) and relatives (52%, 251/486) also supported consent from a doctor when known relatives could not be reached, although slightly under half of practitioners (46% 264/582) supported consent from a doctor in this situation. Noting the contributions of research in informing scientific knowledge and the provision of clinical care, many participants were in favour of consent from a doctor on the proviso that 'research is doing no harm and it's only doing good, then, in my opinion, a doctor should be able to make that decision' (Relative 132). Nevertheless, in all three stakeholder groups approximately 25% of those surveyed disagreed with the concept of consent from a doctor for ICU studies whatever the situation.

A concern that underpinned the views of many was that doctors might prioritise meeting recruitment targets over the welfare of the patients and therefore include patients in their studies who might not be suitable.

> They might have a different interest in the study than, than the patient, hopefully not but you, you just don't know do you. I mean, I think there's a bit of a, erm, old-fashioned view possibly about, I don't know, money, drug companies wanting to push products, who knows and maybe doctors have a bit of, you know, they've got budgets and financial restraints.
>
> Relative 603

> The way that I see that would be like my job. I'm a mechanic, it would be like me doing say a service on my own car, if I was doing a service or an MOT on another customer's car, I'd be like proper critical of it whereas if it was my car, maybe I wouldn't be as critical because it's my car. So maybe, that's like maybe a doctor would be I'm doing this study, right I'm pretty sure she'll be fine whereas if it's a doctor that's not doing the study, they might be a bit more you can't take the blood because… .if you see what I mean.
>
> Relative 492 (group interview)

However, interviews indicated that stakeholders were generally more flexible towards doctors consenting on behalf of incapacitated patients than their survey responses suggested. When interviewed, several participants who had previously disagreed with consent from a doctor in the survey expressed views that were markedly more favourable, although depending on particularities of the study and situation.

> I mean, doctors have to act in your best interest, so they should always put your welfare first, and if the research isn't going to damage that then I haven't got a problem with it, no.
>
> Relative 1532

At interview, regardless of their responses on the survey, participants tended to deliberate over the acceptability of consent from a doctor. Participants whose survey responses indicated a disagreement with consent from a doctor in relation to at least one of the above scenarios felt that these options did not take the patients' wishes into consideration.

> I'd be very reluctant to give that as a, um, as a way of giving consent. Um, doctors vary just like the rest of us, um, and to say that somehow doctors are gifted with, um, some sort of insight into people's wishes, er, I think would be a very rash thing, rash thing to do.
>
> Patient 703

> I think it's, in principle, nothing wrong with it but there's got to be the ethical consideration of how someone's likely to feel about it and again, that could come into the type of study it is.
>
> HCP 4111

Most participants were reassured when the interviewer explained that doctors serving as consultees must be independent of the research study because 'in some ways it might be better, because he's giving an, an absolutely free, er, decision, isn't he?' (Relative 805).

Other participants suggested that it was ethically questionable to use consent processes that risked denying patients the opportunity to be involved in research. They felt that consent from a doctor helped to avoid restricting research progress. Most also seemed more comfortable with doctors giving consent if safeguards were in place and the patient and/or relatives were consulted as soon as practicable after study enrolment.

> Yes, I think as soon as the patient's able to give consent, they need to be approached. Um, but I also appreciate quite often when it comes to somebody who's in ICU, they have been unable to give consent in any way shape or form. Um, so I think obviously as soon as somebody can be approached about something like that, they have to be because, as I said previously, it's personal choice
>
> Patient 713

### Understanding the concept of ICU research

Though research and care are interrelated in the ICU, as we note above, many patients and relatives were confused about the distinctions between the two. Most participants believed that joining a trial would almost certainly benefit the patient.

> But with them asking that, you know, 'go away and think about it', again me and my sister were having a discussion of, obviously they must think that he can help in some way, it must be a positive sign. So we actually didn't need time to think about it, it was, right let's go for this.
>
> Relative 132

> He asked, like, if everyone agrees… would your mum want it done, um, and we said yeah, 'cause obviously if there's summat going that's gonna help her, she'd, she'd take it. So we said yeah, and then they started her on the machine.
>
> Relative 235

Some interviewed patients and relatives suggested that they viewed research on treatments within the ICU (eg, comparing the effectiveness of two widely used treatment strategies) as similar to early phase drug trials for patients for whom standard treatment options have been exhausted. During interviews, the researcher introduced the concept of a phase 3 trial (which are more usual in ICUs than early phase trials), for example, a study comparing different doses of an established treatment, or a new way of giving an established treatment. While about half of participants asked were accepting of such research, others favoured receiving standard care, as '[i]f something's well-proven, it ain't broke, don't fix it, you know' (Patient 711). This preference for standard care was more pronounced when relatives were approached at a particularly difficult time, with some specifically commenting that it was unfair to conduct research involving a change from standard care when a patient is very ill or likely to die, as they were concerned that it may cause the patient discomfort.

> Like if it's changing medications and stuff, I think that's sort of unfair, because if there is risks it could put them in a great deal of pain, especially being towards the end of their lives.
>
> Relative 1218

Conversely, other participants felt that enrolling a patient in research was more justified when they were particularly ill or close to death:

> You know, if your chances of survival are very slim, then, yeah, go for it, because that's the only way that you'll ever develop new technology. If your chances of survival are very high, why would you take the risk?
>
> Patient 711

Some patients and relatives were initially surprised to hear that research occurred in the ICU at all, and had expected that patients were too ill to be involved. On reflection during the interview, participants understood the necessity of research in this context.

> I'd never thought about it, and I was actually mildly surprised, because it's such a weird environment, I wou-, I wouldn't have thought that they'd let researchers in, I thought they'd leave people alone; which I know is stupid, because you have to try things…
>
> Relative 2025

### Perception of risk in research

Many stakeholders noted that factors such as the invasiveness and the risks of the study could modify their views.

Procedures perceived as 'invasive' were usually those that participants felt could cause discomfort to the patient. Participants felt such procedures should be avoided as much as possible as they might increase the risk of harm to a patient (although a study did not necessarily need to involve procedures perceived as invasive for it to be considered risky). The greater the perceived invasiveness or risk, the more participants felt relatives should be involved in the decision about research. However, participants' perceptions of invasiveness and risk differed. For example, a minority of participants were concerned that phlebotomies and taking blood samples may be harmful for a critically ill patient.

> The testing wasn't going to be done on swabs or blood samples that he was going to be having taken for other reasons; it was going to be additional bloods and additional swabs that would be required, and therefore I felt it could potentially have a risk factor for introduction of a, of infection… that wasn't a risk we needed to take at that time.
>
> Relative 340

> She [other relative of patient] felt that it could have harmed him by taking the blood, even though intellectually… she did know that that would not be the case, but I think it, it was an emotional response, because …she knows what rules you have, and that they're not gonna take any blood out of a critically ill patient if it's going to harm them, but I th-, think emotionally…
>
> Relative 225

Most other participants used phlebotomies as an example of a procedure that was either minimally invasive or not invasive at all, and therefore saw these as carrying little risk.

> A blood sample, given that, you know, assuming that it's not gallons and gallons and gallons of blood, um, then I think there's a n-, yeah, I think that's fine, because, I mean, I'm not a science person, so… but I'd assume that won't have too much of a material impact.
>
> Relative 524

Participants' perceptions of risk were often influenced by their beliefs regarding the patient's vulnerability, such as their age and the severity of their condition. For example, the relative of an older patient who was already undergoing an extensive number of invasive procedures believed that the patient was particularly likely to be adversely affected by research that involved additional blood sampling.

> Because patients have… blood tests every four hours, …and so, like when you're sitting there with a relative, you see them constantly taking out little tubes of blood for the blood tests -and it adds up a bit…- you now 150ml is, for a healthy person it's not very much,

but for someone who's really sick and weak, I thought it's too much.

> Relative 2252

Many HCPs interviewed commented that risk is difficult for patients and relatives to understand, and challenging for HCPRs to communicate.

> I'm not sure how well the population understands the concept of risk. I think, as clinicians, we struggle to understand the concept a lot, um, and we deal with it every day. I don't think people necessarily do understand the difference between a risky study and a not risky study, and that's even if we quote them specific figures of rates of complication.
>
> HCP 206

At interview, participants suggested that patients or relatives may misunderstand information about research, or the implications that research may have on the patient's outcome. For instance, some relatives explained that they were unclear how much blood would be taken for a study and one mentioned having declined a study over concerns that the quantity to be taken would harm the patient.

> These are the risks that I in my head believed would have come if I had have agreed yes. Nobody sat down and went through the risks, um, with me… when I said that that was my reason for declining, they didn't disagree or convince me that that wasn't the case.
>
> Relative 340

> We were confused about the amount of blood, but everything's confusing. Even the car park machine's confusing then, … it's a, a lot to, to take in.
>
> Relative 225

HCPRs expressed reluctance to explore and correct such misunderstandings, as they were concerned that this may make a person feel pressured to consent. Conversely, patients and relatives were in favour of having misunderstandings or misinformation corrected, providing that there was no expectation that they agree to participate, although they noted that corrections should not be made when a patient or relative is visibly distressed.

> I think it's acceptable to ask the patient's family, you know, ask why they don't want to, er, to enrol, er, their family member in the research project. I think it's fine, and if they say, oh, because you're taking a gallon of blood, um, say, actually, no we're not, it's just… one, er, syringe… I think that's fine.
>
> Relative 5251

### Perceptions of evidence base

Participants' views were also influenced by their understanding of the evidence base for medical treatments. Survey responses indicated that stakeholders had little awareness of the weak evidence base for treatments that are routinely used in the ICU. Eighty-five per cent

(278/328) of patients and 87% (421/485) of relatives surveyed agreed with the statement that all treatments given to patients were thoroughly tested through studies. At interview, when introduced to the possibility that not all treatments are tested through research, many patients and relatives expressed concern that this meant that patients were being experimented on or used as 'guinea pigs' without consent. Sixty-three per cent (366/582) of HCPs also believed that all treatments given to patients were thoroughly tested through studies. When broken down by role, it was clear that it was nurses with no research duties who were agreeing with this statement (73%, 298/410); in comparison just 21% (8/39) of doctors with research duties. At interview, nurses with no research duties expressed similar concerns as patients and relatives, whereas doctors with research duties discussed the variety of situations where a treatment may not have been researched thoroughly, or researched within the ICU itself:

> A lot of things have not had thorough, erm, er, A-class research to prove that it's used in ICU… That's my understanding and I think it's stuff that, something that consultants talk about a lot. There's old textbooks that we're still going by for using certain protocols and, erm, and they've never really sort of been fully investigated
>
> HCP 404
>
> I know that a lot of the treatments that we do, um in not just intensive care but in medicine and certainly in nursing um have not been tested through rigorous clinical trials because it's not been possible to do that.
>
> HCP 102

### Collaborative decision-making

While consent by a relative is distinct to consent from a doctor, interviews pointed to the possible merits of establishing a collaborative approach to consent in the ICU that incorporates elements of consent by a relative and by a doctor. In interviews, relatives often cited being unsure of how the patient would respond to the risks and benefits of a particular study. Some felt that responsibility for consenting patients to studies should lie with doctors, or were themselves reluctant to take this responsibility, because relatives are 'not really the best people to be making that decision unless they're doctors themselves' (Patient 4280). Others commented that relatives and patients would typically lack the required medical knowledge to appreciate fully the implications of a specific study, yet they still felt that the final decision should remain with the patient or relative.

To explore possible resolutions of this, the interviewer asked participants about possible alternatives to existing procedures that would enable patients and relatives to feel more comfortable and confident in deciding to consent to research. Several participants spoke about the potential for discussions between patients/relatives and HCPs, and a collaborative approach to decision-making

to be helpful in these circumstances. For example, discussion could involve a member of the clinical care team (typically a doctor who is independent of the study, and who might otherwise serve as a professional legal representative or nominated consultee) supporting relatives by responding to their questions or concerns about how a particular study might affect a particular patient. The decision to consent to research or not, would remain with patients or their relatives.

> I think situations I've come across a collaboration approach would actually benefit people. Yeah, I think it would. Why haven't we done it before?
>
> HCP 805
>
> If you did that sort of in collaboration with patients, um, and relatives, like when you're making it, then you could make it so that it was understandable and that, um, it was sort of like well received by people.
>
> HCP 447

### Consent over the phone

Survey responses indicated that when a patient is too ill to decide for themselves, most patients (64%, 329/333), relatives (76%, 487/488) and practitioners (59%, 584/588) considered it acceptable for a doctor to ask a relative over the phone for an opinion on whether the patient should be included in a research study. At interview, most participants explained they supported obtaining consent over the phone when the alternative was a doctor providing consent, as this ensured that the decision remained with relatives.

> It depends on the situation … I think they should try and, and get consent from family, even if it's over the phone, um I think it's, it's better, because… that somebody who knows the patient is doing that, um, and somebody who's a bit more objective to weighing up the risks.
>
> Relative 502
>
> It's probably okay to do it over the phone, erm, if you've got enough, you know, time to make that phone call a valid one… maybe it needs to be a kind of planned phone call … so that you, erm, you are able to take that call in the right place… I don't live anywhere near the hospitals that we've been involved with. So for me it could have been very difficult not being around or just having a quick visit, you know. So I think, yeah, to do it over the phone, there's potential for that to be an acceptable way.
>
> Relative 606

Other participants, primarily HCPs, criticised this approach because staff are unable to gauge body language to tailor their approach to the relatives, and were concerned that relatives might 'feel they have to decide there and then' (HCP 112), although some HCPs valued the option if a relative lives far away or is unable to come to the hospital to give consent:

You've got somebody who perfectly fits your criteria um and there is the logistical hurdle of somebody's living in America who is their next of kin… it would be wrong not to contact them just because they live so far away and, you know, because it's not ideal. I still think it would be, that that should be done and it shouldn't be a barrier to um trying to recruit people.

HCP 102

A few HCPs were also unsure how to formally record consent over the phone while others highlighted the impracticality of asking someone to consent to a study when they would not have an information sheet to refer to. Patients and relatives also emphasised that receiving a telephone call while a relative is in the ICU can be distressing as 'maybe I'd been a bit conditioned that every phone call from a hospital is awful' but noted that it would be acceptable to approach a relative about research over the phone if HCPs 'were to start off by saying, [NAME], sorry for calling, it's not bad news but we haven't seen you today, do you mind, of course' (Relative 144).

Overall, many participants indicated that phone calls were 'better than nothing' (Patient 196) but should only be used when there is a tight time frame and/or the relatives are not physically able to go to the hospital to provide consent, and that the study might otherwise be jeopardised.

### RWPC in the emergency context

Many participants initially struggled to understand a situation where research needed to occur urgently with no time to approach relatives for consent. Some participants were concerned that a RWPC process meant that the research was being conducted without anyone knowing about it.

I don't think I'd like the idea of them just doing something for research without anybody knowing about it. I don't think I'd like that. The other way round, if research is done, I mean, it brings to mind the case of Alder Hey, many years ago.

Relative 113

However, after further discussion at interview and discussion of situations when RWPC was used, many patients and relatives acknowledged its necessity and were more comfortable about the use of RWPC, provided the family would be subsequently informed.

That's not too bad because at least you've got a reason for doing it and you've spoken to the person and the family or whoever later and then told them what's happened, explained what's happened and if they're happy with it and offered to remove it if they insist. I, I think that's acceptable.

Relative 1404

### Informing bereaved relatives of research

Of patients who enter ICUs, 25% do not survive, and therefore, in this context there will be patients enrolled in studies under an emergency consent process or via consent from a doctor who die before the study is discussed with the family. In such circumstances, HCPs were concerned that relatives might misunderstand the situation and believe that the study had contributed to the patient's death, and so 'We leave it…in most intensive care studies we'll have it written that you don't need to approach [a bereaved relative].' (HCP 514).

Ten patients and relatives were introduced to this scenario when interviewed, with the researcher explaining that we were referring to situations in which there was no indication that a study had an influence on the patient's death. When asked whether they felt bereaved relatives should be told about a patient's enrolment in a study, most patients and relatives expressed a preference for being told.

R1 I think for transparency, you have to tell them…

R2 Honesty's the best policy.

R1 …'cause why wouldn't you tell someone? I know it's… might be because you don't want to upset them at that time, but I think at some point they could be told, 'cause I think it would be wrong not to, yes. Why wouldn't you?

Relative 1531 and Relative 1532 (group interview)

Some people would probably say that you know the patient's deceased, there's no point in distressing the family even more. Erm, but I would think it would just be- it's a difficult one, but I would lean towards telling them. I think for one, I think to start off, courtesy, and two kind of similarly echoing the previous scenario, knowing that their loved one has passed, you know, contributed to some research could give them kind of a positive from it, erm, which could help them through that process.

Relative 1189

The one exception was a participant who felt there was little to be gained by informing relatives:

I don't think there's any benefit really in telling the family. I think, um, providing that, you know, the death wasn't a result of the research being undertaken, then … putting myself in them shoes, I wouldn't be too fussed whether or not, you know, my dad took part in a research project or not if he hadn't made it

Relative 524

While patients and relatives were largely in favour of disclosure, they differed in their views of the timing of such disclosure. Some participants preferred to know as soon possible, whereas others believed disclosure would be more appropriate after some time has passed. A few were unsure and acknowledged the complexities of timing the disclosure.

My initial reaction was, well I don't think it would be a very good idea when the person has just died. On the other hand, I don't think I would have been very impressed if I was told several weeks later

Bereaved relative 141

Participants also acknowledged that such disclosures may lead the relatives to believe that the study contributed to the death of the patient, but felt that HCPRs should address these concerns during the discussion.

## DISCUSSION

Little research has explored stakeholders' perspectives on recruitment and consent processes for ICU studies in Europe, and evidence is particularly lacking in the UK. Our findings illustrate the variety of views about consent processes in ICU studies and largely support existing guidance and regulation governing UK research in this area. However, our findings also raise important caveats, particularly where stakeholders hold conflicting views (including within the same stakeholder subgroup), and also provide insights into how HCPRs address issues relating to therapeutic misconceptions, misunderstandings about research studies, handling bereaved relatives of research participants, and reveal variations in practice. Based on the findings, we suggest adaptations to existing procedures that could more closely align with the perspectives of multiple stakeholders. These form our good practice guidance for the recruitment and consent of patients in the ICU.[46]

### Findings endorsing current practice

Mirroring current regulation, guidance and practice, participants preferred informed consent to be sought from patients where possible. In the likely situation where ICU patients lack the capacity to consent for themselves, most participants believed it was acceptable to use SDMs, particularly patient relatives. Such preferences were largely based on the perception that relatives often know the patient well enough to reflect their views. There is evidence that relatives often hold inaccurate understandings of patient wishes, though this evidence is inconsistent.[49] Though participants were broadly positive about consulting with relatives or seeking consent over the phone, findings on this point to the importance of ensuring the process is well managed, that studies are well explained, and relatives' responses formally recorded. With some important caveats, participants also generally accepted the current guidance and practice around the use of doctors serving as SDMs.

Our findings concur with evidence from outside the ICU that some HCPRs find the process of seeking consent particularly challenging.[3 19 50 51] Our study demonstrates that this also applies to ICU research, and reiterates the importance of—and difficulties around—clear, consistent and sensitively communicated information at each stage of recruitment and consent. Participating patients

and relatives were primarily concerned about the impact of any research procedure on a patient's condition and chances of recovery. A few indicated that they considered a patient's involvement in research to be too much of a risk, and consequently, that they would decline to participate in any study. As other research has shown, relatives may disagree about whether a patient should take part because they are anxious and concerned that participation will pose significant risks to their health.[52] HCPs may put patients/relatives at ease by emphasising that their priority is the care of the patient.

### New insights in the ICU research setting

Findings illustrate some divergence in practice and disagreement among HCPs, regardless of their research experience, as to whether it is appropriate to correct misunderstandings relating to research. Patients/relatives expressed uncertainties around practical implications of research for the patient, associated impact on patient outcomes, and implications of diverging from 'standard care'. We have also highlighted some problems with communication between HCPs who are not directly involved in research and those who are. Circumvention of this is important to ensure acceptable and effective recruitment and consent in the ICU.

Our findings illustrate how bedside nurses can serve as trusted gatekeepers to ICU patients and SDMs, and help provide context of the patient's condition, contribute to a culture supportive of research in general and, perhaps, highlight the time-sensitive nature of some research. The importance of bedside nurses in facilitating recruitment has been documented elsewhere.[10 53 54] Other studies have emphasised the role clinical teamscan play in this.[10] Our findings demonstrate how different team members can inform and facilitate consent and recruitment, and help patients/relatives to feel supported.

In addition, we have shown that the views of patients and relatives are often context-dependent. In our study, disagreement with the concept of doctors consenting to research on behalf of patients typically arose from participants being unaware of the safeguards in place to avoid conflict of interests in recruitment, namely that they must be independent of the study.

### Potential room for improvement

Our findings confirm that patients and relatives value clinical information before being asked about research. However, this was not always done consistently or effectively. Timing the approach appropriately and providing an update on the patient's condition before raising research will help to convey that the patient's care is the priority and maintain a sense of trust. Patients and relatives can then be in a better position to understand the broader potential implications of the study, such as benefits to others, allowing the patients/relatives to contextualise what is being asked of them to their unique circumstances.

We found that patients/relatives are amendable to a HCPR exploring and correcting any misunderstandings, provided there was no expectation that this would change their decision. Nevertheless, we note that studies of research consent outside the ICU context indicate that HCPRs are reluctant to explore understandings and correct misunderstandings, with some regarding this as coercive.[55 56]

Patients and relatives would better understand study procedures and risks if researchers explained these using familiar examples, such as talking about the amount of blood to be taken in terms of teaspoons rather than millilitres. Where appropriate it may also be helpful to personalise information about potential risk, for example, by discussing the likely impact of taking a teaspoon of blood for a particular patient. Consistency on this could be monitored and encouraged during the REC approval process.

Patients and relatives indicated the value of collaborative discussions with HCPs who are knowledgeable about a patient's condition in supporting their decision-making. This collaborative approach reflects discussions in the ethical literature in the last two decades critiquing individualistic conceptions of autonomy and calling for more relational understandings of autonomy.[57] These calls are potentially gaining ground in research contexts, just as shared-decision making is becoming more widely accepted in clinical practice outside research.[58] In any event, this collaborative approach is not a new consent process, but rather an adaptation to support relatives who might otherwise feel overly burdened by the responsibility of decision-making. A doctor who is independent of the research might be well-placed to support such collaborative discussions, where practicable within the ICU setting. 'Collaborative decision-making' closely resembles the well-established concept of 'shared decision-making', where patients/carers work together to make treatment decisions.[59] However, shared decision-making is typically associated with decisions about treatment, not research. Though treatment and research are of course often connected, the latter will entail additional or alternative considerations, depending on the type of study. We recommend further work expanding on the conceptual differences between shared and collaborative decision-making.

Currently, when a patient is enrolled using emergency consent procedures or via consent from a doctor and later dies, there is no legal obligation to inform bereaved relatives that the patient was enrolled in a study, unless there is reason to believe the death might be linked to the study. Findings from this study mirror those in paediatric critical care studies,[60–62] indicating that relatives want to be informed about the patient's participation in research. However, we were only able to explore this topic in a small number of interviews. More research is required to confirm the disclosure preferences of bereaved relatives and determine the most appropriate method of informing them. We also note that since our data were collected, the COVID-19 pandemic has normalised research participation to some extent, with increased public awareness of the importance of research,[63] and of the continuum between clinical care and research.[10]

## Limitations

First, to ensure the survey was not overly long, we did not collect sociodemographic data on survey participants, with the exception of age and gender. Recent research suggests that different ethnic minority groups may have different views about consent procedures,[64] which warrants further exploration. Second, while we encouraged participating sites to complete screening logs to enable us to report survey response rates, most sites did not have the staffing to support this, and we are therefore unable to report response rates for the survey.

At interview, exploration of certain topics was limited with some participants. During qualitative data collection we identified the value of incorporating questions about approaching bereaved relatives regarding research after RWPC or consent from a doctor. The interviewers felt the issue was too sensitive and potentially upsetting to raise with some participants. Future research is needed to further explore stakeholder perspectives on this important issue.

Participants also varied in their levels of understanding of certain concepts, both with respect to research and ethical considerations, which sometimes necessitated considerable discussion and explanation during interviews. Consequently, this limited exploration of topics with some participants. It is also important to note that some participants had first-hand experience of being approached about research when in the ICU, and others did not. Therefore, many interview responses regarding the appropriateness of different consent and recruitment practices were based on hypothetical reasoning based on experiences in the ICU, and not direct experience of being approached about research.

**Author affiliations**
[1]Department of Childhood, Youth and Education Studies, Manchester Metropolitan University, Manchester, UK
[2]Department of Public Health, Policy and Systems, University of Liverpool, Liverpool, UK
[3]East and North Hertfordshire National Health Service Trust, Hertfordshire, UK
[4]School of Health and Social Work, University of Hertfordshire, Hatfield, UK
[5]Centre for Social Ethics and Policy, The University of Manchester, Manchester, UK
[6]Department of Biostatistics, University of Liverpool, Liverpool, UK
[7]Department of Critical Care, Liverpool University Hospitals NHS Foundation Trust, Liverpool, UK
[8]Institute of Life Course and Medical Sciences, University of Liverpool, Liverpool, UK
[9]Ulster Hospital, Belfast, South Eastern Health & Social Services Trust, Belfast, Ireland

**Acknowledgements** Authors are grateful to the participants of The Perspectives Study. We are also grateful to the Perspectives Study Advisory group, listed below. Mike Ross, Steve Dilworth, Anthony Austin, Juliet Austin, Alan Brown, Stephen Brett, Kathy Rowan, Tim Walsh, Angus Dawson, Tim Coates, Keith Young, Harriet Treare, Clive Collett, Catherine White, Ron Daniels, Sue Dean, Paul Dark, Heather Rogers, Kate Neville, Beth Hazel, Paul Mouncey, Marlies Ostermann, Laura Ortiz-Ruiz De Gordoa, Joanne Outtrim, Karen Williams, Christina Jones, Sara Campos, Neus Grau,

Valerie Page, Johnathan Bannard-Smtih, Phil Hopkins, Sonya Finucane, Owen Boyd, Richard Pierson, Julian Sonkesen, Richard Innes, Annette Woods, Jane Sansom, Sarah Williams

**Contributors** BY, KW, LF, CG and IW designed the Perspectives study. KP and AK conducted the interviews and analysis of quantitative data. KP led on the analysis of qualitative data. KP, BY, KW, LF, CG, IW, NP, JT and AK contributed to the data analysis. LF provided insights from empirical bioethics. BY is the guarantor for this paper. KP, BY, KW, LF, CG, IW, NP, JT and AK read and approved the final manuscript.

**Funding** The Perspectives Study was funded by the Economic and Social Research Council ES/N006372/1.

**Competing interests** None declared.

**Patient and public involvement** Patients and/or the public were involved in the design, or conduct, or reporting, or dissemination plans of this research. Refer to the Methods section for further details.

**Patient consent for publication** Not applicable.

**Ethics approval** This study involves human participants and was approved by North West - Liverpool Central Research Ethics Committee: 17/NW/0339. Participants gave informed consent to participate in the study before taking part.

**Provenance and peer review** Not commissioned; externally peer reviewed.

**Data availability statement** Data are available in a public, open access repository. The datasets generated and/or analysed during the current study are available in the UK DataService repository, https://reshare.ukdataservice.ac.uk/854286/ and https://datacat.liverpool.ac.uk/1510/

**ORCID iDs**
Katie Paddock http://orcid.org/0000-0003-1264-5845
Lucy Frith http://orcid.org/0000-0002-8506-0699
Ingeborg Welters http://orcid.org/0000-0002-3408-8798

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
