## [Reviewer comments · BMJ Open]

ARTICLE DETAILS

TITLE (PROVISIONAL)	Learning from stakeholders to inform good practice guidance on consent to research in intensive care units: a mixed methods study
AUTHORS	Paddock, Katie; Woolfall, Kerry; Kearney, Anna; Pattison, Natalie; Frith, Lucy; Gamble, Carrol; Welters, Ingeborg; Trinder, John; Young, Bridget

VERSION 1 – REVIEW

REVIEWER	Shepherd, Victoria Cardiff University, Centre for Trials Research
REVIEW RETURNED	15-Jul-2022

GENERAL COMMENTS	Thank you for the invitation to review this very interesting manuscript reporting a study to explore the perspectives of different stakeholders towards consent procedures in research conducted in ICUs. The authors write on an important topic that will be of interest to those involved in designing and conducting critical care research, as well as other stakeholders such as members of research ethics committees. The manuscript is well written and enjoyable to read, and I congratulate the authors on conducting a valuable study. However, there are a few areas of the manuscript that would benefit from revision in order to strengthen it, particularly around what constitutes the good practice guidance. Introduction p.5 As the legal arrangements for the involvement of substitute decision-makers (and the associated terminology) differs between jurisdictions, it would be helpful to include that and add references to the legal frameworks and therefore terminology used in the paper (i.e the examples of SDM are specific to England and Wales rather than generic). Suggest that the term 'professional consultee' should be 'nominated consultee' or expanded to incorporate a professional legal representative. Additionally, a doctor is only one example of who might be nominated to act as a consultee for the purposes of section 32 of the Mental Capacity Act (see DoH Guidance on nominating a consultee for research involving adults who lack capacity to consent) or professional legal representative under Schedule 1 Part 5 of Medicines for Human Use (Clinical Trials) Regulations (see HRA Consent and Participant Information Guidance). p.5 lines 31-35 Suggest revising the wording for clarity of 'Implementing substitute decision making also has its challenges, particularly those due to the short time windows for recruitment' –
---

	perhaps 'particularly where there is a short time window for recruitment' p.6 line 40 'HCPs' should be written in full for this first use Methods Survey – more details are needed here about the survey response options (later on there is reference in the analysis section to Likert-type - and was there a neutral option (e.g neither agree nor disagree)?) Interviews - later in the paper (Results section p16) there is a quote attributed to a relative from a group interview. It wasn't clear in the Methods section that interviews were not all one-to-one. Additional detail is needed if that is the case. Analysis – given that descriptive statistics are used why were Likert responses combined for strongly agree and agree, and strongly disagree and disagree? It is not easy to get a sense of the survey findings and range of responses without more quantitative data being presented. p.7 line 40 the NIHR has recently been renamed to the National Institute for Health and Care Research p.9 line 50 reference is missing (REF) Results There is no mention in details of the survey participants nor interview sample about the nature of the relationship for relatives of ICU patients. Given the relationality impacts on substitute decision-making, this should be reflected on in the manuscript (e.g limitation if no data were collected on relationship, relevance of this to the findings in the discussion). P.12 refers to the terminology used in the HRA guidance but this reflects the MCA and Clinical Trials Regulations so potentially these should be considered the source of the terminology and referenced. It also again refers to nominated consultees being doctors which as previously commented is only one example. P.13 line 21 has an extraneous '2' at the end of the line p.17 lines 29-31 In the sentence about 'using a tool to assess capacity formally' it wasn't clear whether the HCPRs were using a specific tool and being asked about their experience of it or if they were being asked if they use/used a tool. Clarifying that sentence would be helpful. p.20 In the title of the section, and from then on throughout the text, the term 'professional consent' is used. This term is problematic as it is intended to mean 'consent by a professional' but implies that the consent is professional – as contrasted with implied amateur or unprofessional consent given by family members. The term is sometimes used in practice settings but I would encourage the authors not to use this term both for this reason and because it doesn't reflect the legal terminology. P.20-21 I would encourage the authors to reflect (in the results and/or discussion section) on relatives' concerns that doctors
--	---

	might prioritise meeting recruitment targets over the welfare of the patients, and if this might have been influenced by (mis)understanding about the independence of the doctor as consultee/legal representative - or as highlighted it might not be a doctor. E.g referring to it being 'their studies'. The authors do go on to say that most participants were reassured when the interviewer explained that doctors serving as consultees must be independent of the research study, but there are no quotes to illustrate/reinforce this important point and contextualised the survey findings that 25% disagreed with 'professional consent'. p.30 It wasn't clear how collaborative decision-making might differ from shared decision-making, or even supported decision-making – both of which are well-established concepts. Expanding on the conceptual differences would be helpful here or in the discussion section. Also, how might a member of the clinical care who is independent of the study be best placed to answer questions or concerns about how a particular study might affect a particular patient, compared to an investigator who is familiar with the study? p.31 The term professional consultee is used again here – see previous comment Discussion P.36 The first sentence of the discussion is repeated from the introduction and is repeated again at the end of the discussion section. Suggest removing the duplications. A relatively low percentage of survey participants had been approached about research or were involved in conducting research, and less than half of the patients interviewed had experience of being approached to provide consent. Whilst this is included in the limitation section, greater consideration could be given to the impact on responses where many are based on hypothetical reasoning and not direct experience. Similarly there is reference to the limited socio-demographic data available, which is well justified, and the earlier strengths and limitations section refers to sampling for diversity. However there is no mention of ethnicity in the manuscript which is an important area to address given the issues highlighted during COVID-19 research, and that different ethnic minority groups may have very different views about the acceptability of consent – particularly in emergency research (e.g https://doi.org/10.1186/s13063-021-05568-z). I am intrigued about the origin of the question “all ICU patients should take part in clinical research studies, unless a doctor advised against it”. I am not sure what contribution that brings – if participation is based on the patient's wishes we wouldn't expect all patients to have positive view about participation. The analysis section refers to empirical bioethics and drawing normative conclusions, but an empirical bioethics approach is not necessarily apparent throughout the manuscript. Perhaps this could be made more explicit and returned to in the discussion if not elsewhere. E.g what are the normative implications of the findings?
--	---

	The title and the analysis sections etc refers to the development of 'good practice guidance'. It is not clear what constitutes the good practice guidance beyond the results. This is an important area to address. If the guidance is woven in the results/discussion then drawing them out, perhaps in the form of a table or list of recommendations, would enhance the impact of the study. There doesn't appear to be a conclusion section although there is one in the abstract?
--	---

REVIEWER	Bitter, Cindy Saint Louis University, Surgery/Emergency Medicine
REVIEW RETURNED	02-Aug-2022

GENERAL COMMENTS	This is an interesting study on a topic that is important and not well researched. The article would be more readable if grammar was checked and sentence structure was simplified. For example, page 5 line 21 - commas not necessary after "(SDM)s," and "incapacitated patients,". Sentence page 5 lines 26-30 is unclear. Page 6 line 40 - define HCPs at first use. Page 7 line 52 define HDU at first use. Page 10 "Each version included, where possible, the same questions, with wording changed to reflect the respondents" could be simplified to "where possible, each version included the same questions, with wording changed to reflect..." Likewise page 14, lines 12- 19. Not the only instances. The numbers for the Supplementary Files are transposed, page 13.
---

VERSION 1 – AUTHOR RESPONSE

Reviewer 1 comments

Reviewer 1 comment: Introduction p.5 As the legal arrangements for the involvement of substitute decision-makers (and the associated terminology) differs between jurisdictions, it would be helpful to include that and add references to the legal frameworks and therefore terminology used in the paper (i.e the examples of SDM are specific to England and Wales rather than generic).

Author response: We have now included (page 11-12) a brief acknowledgement that the legal arrangements regarding SDMs vary across different countries. Further detail would add to the complexity and length of the paper.

Reviewer 1 comment: Suggest that the term 'professional consultee' should be 'nominated consultee' or expanded to incorporate a professional legal representative.

Author response: We feel it is important to keep the language grounded in the terms that interviewees tended to use and this was often "professional consultee"/"professional representative". For accuracy in the introduction, "professional consultee" has now been changed to "professional legal representative" on page 5.

Reviewer 1 comment: Additionally, a doctor is only one example of who might be nominated to act as a consultee for the purposes of section 32 of the Mental Capacity Act (see DoH Guidance on nominating a consultee for research involving adults who lack capacity to consent) or professional legal representative under Schedule 1 Part 5 of Medicines for Human Use (Clinical Trials) Regulations (see HRA Consent and Participant Information Guidance).

Author response: If we have understood this point correctly, the reviewer is referring to situations outside the ICU and what the Mental Capacity Act says about who can act as consultees in these broader contexts. However, to avoid introducing further complexity to the manuscript, and as our focus is specifically research in ICUs where it is usually doctors to act as consultees/representatives, we are inclined to keep what we currently say here.

Reviewer 1 comment: p.5 lines 31-35 Suggest revising the wording for clarity of 'Implementing substitute decision making also has its challenges, particularly those due to the short time windows for recruitment' – perhaps 'particularly where there is a short time window for recruitment'
p.6 line 40 'HCPs' should be written in full for this first use

Author response: Amended, thank you

Reviewer 1 comment: Methods / Survey – more details are needed here about the survey response options (later on there is reference in the analysis section to Likert-type - and was there a neutral option (e.g neither agree nor disagree)?)

Author response: Reference to our use of a five-point Likert scale has been added to page 10.

Reviewer 1 comment: Interviews - later in the paper (Results section p16) there is a quote attributed to a relative from a group interview. It wasn't clear in the Methods section that interviews were not all one-to-one. Additional detail is needed if that is the case.

Author response: This has been clarified on page 13-14

Reviewer 1 comment: Analysis – given that descriptive statistics are used why were Likert responses combined for strongly agree and agree, and strongly disagree and disagree? It is not easy to get a sense of the survey findings and range of responses without more quantitative data being presented.

Author response: This was to ascertain whether participants were broadly positive, negative, or neutral in their views. This has been clarified on page 11.

Reviewer 1 comment: p.7 line 40 the NIHR has recently been renamed to the National Institute for Health and Care Research

Author response: Amended

Reviewer 1 comment: p.9 line 50 reference is missing (REF)

Author response: Amended

Reviewer 1 comment: Results

There is no mention in details of the survey participants nor interview sample about the nature of the relationship for relatives of ICU patients. Given the relationality impacts on substitute decision-making, this should be reflected on in the manuscript (e.g limitation if no data were collected on relationship, relevance of this to the findings in the discussion).

Author response: We collected data on the nature of the relationship between 'relatives' and patients (son/daughter, brother/sister, mother/father, spouse/partner, friend, other). A summary of this data has been added to Supplementary File 2. With the majority of interviewed relatives (and surveyed relatives) identifying as either son/daughter or spouse/partner, this did not inform our analysis.

Reviewer 1 comment: P.12 refers to the terminology used in the HRA guidance but this reflects the MCA and Clinical Trials Regulations so potentially these should be considered the source of the terminology and referenced. It also again refers to nominated consultees being doctors which as previously commented is only one example.

Author response: I have added references for the MCA and Clinical Trials regulations. “Nominated consultee” has been changed “professional consultee”. Though doctors are just one example of a nominated consultee, we feel it sufficient to refer to “professional consultees” in this context.

Reviewer 1 comment: P.13 line 21 has an extraneous ‘2’ at the end of the line

Author response: Deleted

Reviewer 1 comment: p.17 lines 29-31 In the sentence about ‘using a tool to assess capacity formally’ it wasn’t clear whether the HCPRs were using a specific tool and being asked about their experience of it or if they were being asked if they use/used a tool. Clarifying that sentence would be helpful.

Author response: Amended on page 18

Reviewer 1 comment: p.20 In the title of the section, and from then on throughout the text, the term ‘professional consent’ is used. This term is problematic as it is intended to mean ‘consent by a professional’ but implies that the consent is professional – as contrasted with implied amateur or unprofessional consent given by family members. The term is sometimes used in practice settings but I would encourage the authors not to use this term both for this reason and because it doesn’t reflect the legal terminology.

Author response: Though we think it important to maintain the language used by participants in this paper, we do take your point with our use of “professional consent” in the structure and discussion of the paper. This has been amended to “consent from a doctor” throughout the paper, where relevant.

Reviewer 1 comment: P.20-21 I would encourage the authors to reflect (in the results and/or discussion section) on relatives’ concerns that doctors might prioritise meeting recruitment targets over the welfare of the patients, and if this might have been influenced by (mis)understanding about the independence of the doctor as consultee/legal representative - or as highlighted it might not be a doctor. E.g referring to it being ‘their studies’. The authors do go on to say that most participants were reassured when the interviewer explained that doctors serving as consultees must be independent of the research study, but there are no quotes to illustrate/reinforce this important point and contextualised the survey findings that 25% disagreed with ‘professional consent’.

Author response: We have made reference to the consideration that views on ‘professional consent’ are linked to misunderstandings/uncertainties around doctors being independent of the study (page 23 in the results and 39-40 in the discussion). There are quotes on page 21-23 that illustrate concerns around ‘professional consent’ to understand the finding that 25% disagreed with this (Relatives 603 and 492, Patient 703, HCP 4111).

Reviewer 1 comment: p.30 It wasn’t clear how collaborative decision-making might differ from shared decision-making, or even supported decision-making – both of which are well-established concepts. Expanding on the conceptual differences would be helpful here or in the discussion section.

Author response: We have added the following to page 41 of the Discussion

“‘Collaborative decision-making’ closely resembles the well-established concept of ‘shared decision-making’, where patients/carers work together to make treatment decisions. However, shared decision-making is typically associated with decisions about treatment, not research. Though treatment and research are of course often connected, the latter will entail additional or alternative considerations, depending on the type of study.”

We also recommend further work expanding on the conceptual differences between shared and collaborative decision-making.

Reviewer 1 comment: Also, how might a member of the clinical care who is independent of the study be best placed to answer questions or concerns about how a particular study might affect a particular patient, compared to an investigator who is familiar with the study?

Author response: Our findings suggest that independence of the healthcare practitioner in relation to a study takes precedence over knowledge of the study. Practically, doctors who are not involved in the study would be able to familiarise themselves with it, or seek advice from a study colleague if a question comes up that they cannot answer. But the reverse is not true – a doctor who is involved in a study cannot be independent of it.

Reviewer 1 comment: p.31 The term professional consultee is used again here – see previous comment

Author response: Amended

Reviewer 1 comment: Discussion

P.36 The first sentence of the discussion is repeated from the introduction and is repeated again at the end of the discussion section. Suggest removing the duplications.

Author response: The duplicate sentence has been removed from the end of the Discussion section. Given the length and detail of the paper we feel keeping in the repeated point from the introduction to the discussion helps to orientate the reader.

Reviewer 1 comment: A relatively low percentage of survey participants had been approached about research or were involved in conducting research, and less than half of the patients interviewed had experience of being approached to provide consent. Whilst this is included in the limitation section, greater consideration could be given to the impact on responses where many are based on hypothetical reasoning and not direct experience.

Author response: We have edited page 43 to emphasise the role of hypothetical reasoning in the responses of some interview participants. We believe this is sufficient acknowledgement of this consideration as though some participants did not direct experience of being approached about research, responses were still informed by their unique experiences in the ICU.

Reviewer 1 comment: Similarly there is reference to the limited socio-demographic data available, which is well justified, and the earlier strengths and limitations section refers to sampling for diversity. However there is no mention of ethnicity in the manuscript which is an important area to address given the issues highlighted during COVID-19 research, and that different ethnic minority groups may have very different views about the acceptability of consent – particularly in emergency research (e.g <https://doi.org/10.1186/s13063-021-05568-z>).

Author response: We have amended the earlier strengths and limitations section to clarify that our sampling strategy included sampling for diversity of views rather than ethnicity. The later limitations section (page 42) has also been amended to include reference to the importance of exploring differences in views/experiences of different ethnic minority groups.

Reviewer 1 comment: I am intrigued about the origin of the question “all ICU patients should take part in clinical research studies, unless a doctor advised against it”. I am not sure what contribution that brings – if participation is based on the patient’s wishes we wouldn’t expect all patients to have positive view about participation.

Author response: Question design involved a combination of using findings from an earlier workstream of the study (referenced), advisory group feedback, other literature, and piloting multiple versions of the questionnaires across different sites. Adding more detail to this and other questions would have made it difficult to answer, though we do acknowledge that this question received differing interpretations post-pilot. We refer to the piloting and refining of questions on page 10 of the manuscript.

Reviewer 1 comment: The analysis section refers to empirical bioethics and drawing normative conclusions, but an empirical bioethics approach is not necessarily apparent throughout the

manuscript. Perhaps this could be made more explicit and returned to in the discussion if not elsewhere. E.g what are the normative implications of the findings?

Author response: More detail added to this section on page 12 to illustrate how empirical bioethics has been incorporated in to the analysis.

Reviewer 1 comment: The title and the analysis sections etc refers to the development of 'good practice guidance'. It is not clear what constitutes the good practice guidance beyond the results. This is an important area to address. If the guidance is woven in the results/discussion then drawing them out, perhaps in the form of a table or list of recommendations, would enhance the impact of the study.

Author response: The guidance we refer to in the paper is a separate, detailed document for practitioners. We plan to publish an additional paper specifically documenting the guidance in due course. The present manuscript reflects the empirical data that informed the development of this guidance. We have made further reference to the good practice guidance on pages 12 and 38.

Reviewer 1 comment: There doesn't appear to be a conclusion section although there is one in the abstract?

Author response: We have followed the guidelines for abstract structure and other papers published in the journal have included a conclusion section in the abstract but not in the paper. We feel the Discussion section of the manuscript covers the conclusions of the paper also without necessitating a dedicated section.

Reviewer 2 comments

Reviewer 2 comment: This is an interesting study on a topic that is important and not well researched. The article would be more readable if grammar was checked and sentence structure was simplified. For example, page 5 line 21 - commas not necessary after "(SDM)s," and "incapacitatedpatients,".

Author response: This has been amended

Reviewer 2 comment: Sentence page 5 lines 26-30 is unclear.

Author response: This has been amended

Reviewer 2 comment: Page 6 line 40 - define HCPs at first use.

Author response: This has been amended

Reviewer 2 comment: Page 7 line 52 define HDU at first use.

Author response: This has been amended

Reviewer 2 comment: Page 10 "Each version included, where possible, the same questions, with wording changed to reflect the respondents" could be simplified to "where possible, each version included the same questions, with wording changed to reflect..."

Author response: This has been amended, thank you

Reviewer 2 comment: Likewise page 14, lines 12- 19. Not the only instances.

Author response: This has been amended. We have also made minor amendments to wording throughout the manuscript to aid clarity.

Reviewer 2 comment: The numbers for the Supplementary Files are transposed, page 13.

Author response: This has been amended, thank you

VERSION 2 – REVIEW

REVIEWER	Shepherd, Victoria Cardiff University, Centre for Trials Research
REVIEW RETURNED	03-Oct-2022

GENERAL COMMENTS	Thank you for the opportunity to review the revised manuscript. The authors are to be congratulated on carefully responding to the points previously raised which has strengthened the manuscript in preparation for publication. However, there are a couple of remaining comments on the revised manuscript that the authors might wish to consider – primarily around the terminology used, in which accuracy is important given the confusion surrounding their use in practice. For example, the terms ‘nominated consultee’ and ‘professional legal representative’ are sometimes used interchangeably although these are separate terms with specific legal definitions and roles according to the different legal frameworks that apply to CTIMPs and non-CTIMPs. Abstract Thank you for the clarification that the good practice guidance is available elsewhere. It is not clear from the abstract that this is not contained in the paper and so would suggest adding this to the abstract in order to inform readers who may be anticipating that the paper contains the guidance itself. Introduction p.5 lines 26-30 refers to ‘a doctor who is not involved with the study and is therefore considered to be independent, often referred to as a professional legal representative’. Someone who acts as a SDM in a professional capacity may be either a nominated consultee or a legal representative depending on the type of research. This is explained in the Results section but could be moved to the Introduction section to help orientate the reader, and then the difference in use by the public and others is one of the findings. Results p.13 line 24 has been changed from nominated to professional consultees but should remain ‘nominated consultee’ as the legally recognised term. Clarification is needed around the ‘Acceptability of consent from a doctor’ (possibly in the Discussion section) that for non-CTIMPs a consultee (either nominated or personal) does not provide consent on the person’s behalf. Rather they provide advice to the recruiting researcher/investigator who then has responsibility for deciding whether to enrol the person in the study or not. This is an important clarification (even if not part of the survey or interview responses) as it means that the consultee has a consultation role only and the locus of responsibility for inclusion lies with the recruiting researcher/investigator.
---

REVIEWER	Bitter, Cindy Saint Louis University, Surgery/Emergency Medicine
REVIEW RETURNED	14-Oct-2022

GENERAL COMMENTS	Thank you for your work to clarify this manuscript - I believe it reads better after revision. The following need further clarification: Introduction P 5 lines 26-30 - not a complete sentence P 5 lines 42-49 - Please rephrase - does not reflect the clinical equipoise that underpins the need for a study. If we don't know that a treatment will benefit the patient, a delay in administering the treatment should not "compromise patient outcomes". Discussion P 38 lines 40-49 - please rephrase - excessively complex sentence followed by incomplete sentence P 38 lines 49-56 - would put sentence about phone consent after discussion about patients' relatives knowing their views and research that calls this into question.
---

VERSION 2 – AUTHOR RESPONSE

Reviewer 1 comments

Reviewer 1 comment: Thank you for the opportunity to review the revised manuscript. The authors are to be congratulated on carefully responding to the points previously raised which has strengthened the manuscript in preparation for publication. However, there are a couple of remaining comments on the revised manuscript that the authors might wish to consider – primarily around the terminology used, in which accuracy is important given the confusion surrounding their use in practice. For example, the terms ‘nominated consultee’ and ‘professional legal representative’ are sometimes used interchangeably although these are separate terms with specific legal definitions and roles according to the different legal frameworks that apply to CTIMPs and non-CTIMPs.

Author response: Where we refer to ‘professional legal representative’ we have – where necessary – included ‘or nominated consultee’ to clarify this. This occurs on page 5 and 31.

Reviewer 1 comment: Abstract

Thank you for the clarification that the good practice guidance is available elsewhere. It is not clear from the abstract that this is not contained in the paper and so would suggest adding this to the abstract in order to inform readers who may be anticipating that the paper contains the guidance itself.

Author response: This has been added to the Results section of the abstract.

Reviewer 1 comment: Introduction

p.5 lines 26-30 refers to ‘a doctor who is not involved with the study and is therefore considered to be independent, often referred to as a professional legal representative’. Someone who acts as a SDM in a professional capacity may be either a nominated consultee or a legal representative depending on the type of research. This is explained in the Results section but could be moved to the Introduction section to help orientate the reader, and then the difference in use by the public and others is one of the findings.

Author response: We have included reference to nominated consultees also on page 5. We have kept the explanation regarding terminology in the Results section to avoid confusing the introduction.

Reviewer 1 comment: Results

p.13 line 24 has been changed from nominated to professional consultees but should remain ‘nominated consultee’ as the legally recognised term.

Author response: This has been amended, thank you

Reviewer 1 comment: Clarification is needed around the 'Acceptability of consent from a doctor' (possibly in the Discussion section) that for non-CTIMPs a consultee (either nominated or personal) does not provide consent on the person's behalf. Rather they provide advice to the recruiting researcher/investigator who then has responsibility for deciding whether to enrol the person in the study or not. This is an important clarification (even if not part of the survey or interview responses) as it means that the consultee has a consultation role only and the locus of responsibility for inclusion lies with the recruiting researcher/investigator.

Author response: Page 20 – we have included the following clarification, which also links to the explanation regarding terminology at the beginning of the Results section: "As noted above, for non-CTIMPs, nominated consultees provide advice regarding the recruitment of an incapacitated patient to research, not consent on their behalf as they would in CTIMPs[11]. We used the more widely-understood terminology of 'consent' to refer to in the survey and interviews".

Reviewer 2 comments

Reviewer 2 comment: Introduction

P 5 lines 26-30 - not a complete sentence P 5 lines 42-49 - Please rephrase - does not reflect the clinical equipoise that underpins the need for a study. If we don't know that a treatment will benefit the patient, a delay in administering the treatment should not "compromise patient outcomes".

Author response: We have edited this to refer to "investigating under emergency situations" to clarify issues relating to equipoise. Page 5

Reviewer 2 comment: Discussion

P 38 lines 40-49 - please rephrase - excessively complex sentence followed by incomplete sentence

Author response: This has been amended.

Reviewer 2 comment: P 38 lines 49-56 - would put sentence about phone consent after discussion about patients' relatives knowing their views and research that calls this into question.

Author response: This has been amended, thank you.